# Cross-Level Dynamics of Collaboration and Conflict in Multi-Party Systems: An Empirical Investigation Using a Behavioural Simulation

**Petru Lucian Curseu [1,2,\*] and Sandra Schruijer [3,4]**

[1]  Department of Psychology, Babeş-Bolyai University, Cluj-Napoca 400084, Romania
[2]  Department of Organisation, Open University of the Netherlands, Valkenburgerweg 177,
    6419 AT Heerlen, The Netherlands
[3]  Utrecht University School of Governance, Utrecht University, Bijlhouwerstraat 6, 3511 ZC Utrecht,
    The Netherlands; S.G.L.Schruijer@uu.nl
[4]  TIAS School for Business and Society, Tilburg University, Warandelaan 2, 5037 AB Tilburg, The Netherlands
[\*]  Correspondence: petrucurseu@psychology.ro; Tel.: +40-264-590-967

**Abstract:** Multiparty systems bring together various stakeholder parties and their representatives and offer a platform for sharing their diverse interests, knowledge and expertise in order to develop and realize joint goals. They display complex relational dynamics in which within-party interactions (interpersonal interactions within each stakeholder party) as well as between-party interactions (interactions between the stakeholder parties) intertwine to generate bottom-up and top-down influences. We investigate these influences in a behavioural simulation. Our results show that changes in task conflict at the stakeholder party level positively predict changes in perceived collaborativeness in the overall system, while changes in relationship conflict at the stakeholder party level positively predict changes in perceived conflictuality in the system. Moreover, we show that changes in perceived overall conflictuality leads to a proportional change in relationship conflict experienced within the stakeholder parties.

**Keywords:** collaboration; conflict; participation; multiparty systems; group dynamics; multilevel analysis

## 1. Introduction

Complex decisions with major social, environmental and economic consequences are often made by groups consisting of multiple organizations rather than by individuals alone (Curşeu and Schruijer 2017). In multiparty systems stakeholders explore their interdependencies and use their knowledge and expertise in order to integrate and develop their different perspectives and interests (Vansina et al. 1998; Schruijer and Vansina 2008; Schruijer 2016). The decision making in such systems is characterized by high complexity, given the complex nature of the problem domain, the various stakeholder parties involved and the diversities in interests, identities, perspectives and power positions. Relational dynamics in such multiparty collaborative systems shape the decision outcomes (Curşeu and Schruijer 2017; Schruijer 2008). Participants interact within their own stakeholder party to discuss their party's goals, aspirations and interests, while simultaneously they interact with the other stakeholders that have their own, often differing goals, aspirations and interests, so as to arrive at and realize a joint goal. Through this collective goal stakeholders can jointly address a problem which they cannot solve on their own, while through working towards the joint goal, stakeholders can serve their intra-organizational goals (Schruijer 2008).

In such multi-party systems, composed of several stakeholder parties, relational dynamics unfold at two levels: within stakeholder parties as well as between stakeholder parties. These two relational dimensions are not independent as the relational dynamics that arise from the stakeholder parties may influence the larger system as a whole (bottom-up influences) while system-level relational dynamics may spiral down from the larger system to influence the different stakeholder parties (top-down influences). So far, the literature on multi-party systems lacks systematic investigations of these jointly operating influences and it does not explore how task conflict and relational conflict experienced within the stakeholder parties have an impact on the relational dynamics of the system as a whole. Likewise, there is no direct empirical evidence on how changes in conflict and collaboration in the whole system influence the dynamics of task and relationship conflict within the participating stakeholder parties. We set out to explore the interplay of these bottom-up and top-down influences in multiparty systems, using a behavioural simulation.

In line with interdependence theory (Holmes 2002), when people foresee they will engage in social interactions, they build expectations about: (1) the nature of the situation and (2) the goals, motives and behaviours of the ones they will interact with. The expectations about others and the social situation will eventually shape behaviour in a variety of social contexts (Holmes 2002). In other words, in social situations, expectations and social behaviour are entwined.

In field theory and interdependence terms (Lewin 1936; Holmes 2002) two interdependence fields are generated in multiparty systems that explain the relational dynamics within as well as between parties. In the small parties (first interdependence field) people will build expectations about their own party (social situation) and their teammates (interaction partners), while in the larger multiparty system people will build expectations about the interests of other stakeholder parties and the general climate of the multiparty system as a whole. It is our contention that the expectation—behaviour entwinements at the two levels (party and multiparty system) are interdependent. More specifically our paper sets out to explore the extent to which the association between expectations and experienced social interactions within parties influences the expectation-behaviour entwinement in the larger system (bottom up influence) and the extent to which the entwinement between expectations and behaviours in the larger system is tied to the expectation-behaviour association within parties (top-down influence).

We consider the two forms of social interdependence described by Deutsch (1949), namely positive (as illustrated by collaborative intentions and behaviour) and negative (as illustrated by conflictual intentions and behaviours) interdependence. It follows that the entwinement between expectations for positive interdependence and realized collaboration within the stakeholder parties impacts the entwinement between expected collaborativeness and real collaborativeness in the larger multiparty system. Moreover, we also expect that the entwinement between expectations of negative interdependence and experienced (relationship) conflict within stakeholder parties is dependent on the same type of interdependence entwinement at the multiparty system level. We set out to explore the way in which the entwinement of expectations-experiences for positive and negative interdependence is transferred from the parties to the multiparty level (bottom-up influence) and the way in which the expectations-experiences entwinement for positive and negative interdependence at the multiparty level trickles down to impact the parties in the system (top-down influence). This interdependence dynamics and the bottom up and top down influences is depicted in Figure 1.

As the interplay between collaboration and conflict is essential for decision quality in multiparty systems (Curşeu and Schruijer 2017), our study has the potential to make several contributions to the literature. First, we answer the call for dynamic models of the interplay between emergent states in teams and multiparty systems (Costa et al. 2017; Shuffler et al. 2015; Waller et al. 2016) and we use a cross-lagged design to capture changes in collaborativeness and conflictuality. Second, we use a realistic behavioural simulation to tap into the complex dynamics of conflict and collaboration in multiparty systems and to explore the positive role of task conflict for collaboration in multiparty systems. Using a round robin method to evaluate collaborativeness and conflict, we capture both the top-down as well as the bottom-up interplay between collaboration and conflict in multiparty systems.

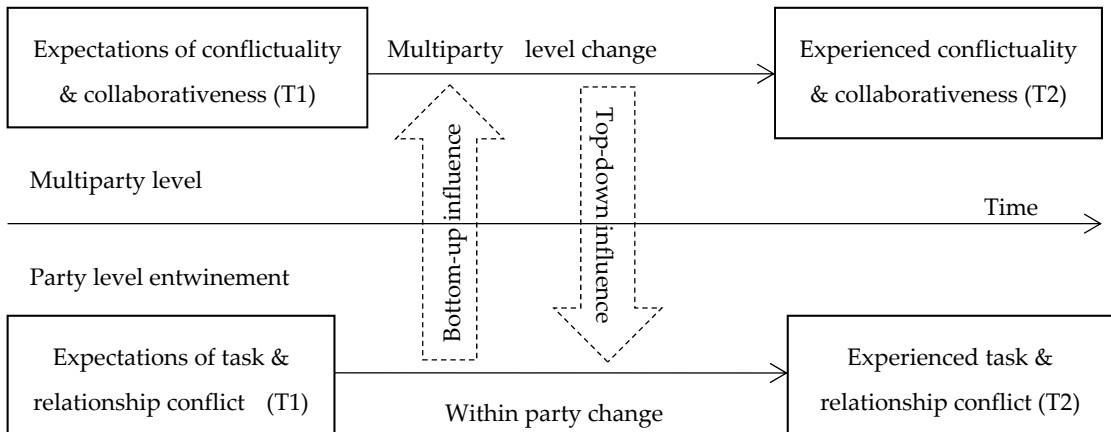

**Figure 1.** Bottom-up and top-down influences in multiparty systems. Note: T1 = time 1 (expectations before the interaction starts), T2 = time 2 (experienced dynamics at the party and multiparty level).

## 2. Theory and Hypotheses

Literature on multiparty collaboration is diverse and lacks integration (Vangen 2017). Theories of multiparty collaboration range from practice oriented theories of collaboration that theorize collaborative relations as ways in which stakeholders can achieve *collaborative advantage* (Huxham 2003; Vangen 2017) to more systemic approaches that distinguish between inputs, processes and outputs of collaborative relationships (Thomson and Perry 2006; Wood and Gray 1991). Multiparty systems are social systems composed of interdependent stakeholder parties that interact to discover ways of moving forward while dealing with existing contradictions (Vangen 2017). Collaborative relationships are inherently paradoxical as the stakeholders have to work with their similarities and their differences simultaneously in order to achieve collaborative advantage (Huxham 2003; Vangen 2017). In terms of organization, multiparty systems are formed of individuals, nested in stakeholder or interest parties that are ultimately nested in a larger system that strives for collaboration. Various interdependencies (both positive and negative) that exist among these systemic components are conceptualized as inputs for collaborative relationships (Wood and Gray 1991). These interdependencies are however not necessarily subject to formal or centralized control and members of multiparty systems have autonomy to discover viable ways of dealing with the paradoxes of collaboration (Thomson and Perry 2006). It is our contention that the paradoxes inherent to multiparty collaboration (Huxham 2003; Vangen 2017) reflect two co-existing forms of social interdependence. We further build on insights from Social Interdependence Theory (Deutsch 1949) to understand how interactions within the stakeholder parties (like conflict) influence the functioning of the whole system (bottom-up influences) while the interdependencies among the parties will shape the functioning of the individual stakeholder parties (top-down influences).

Social interdependence theory (Deutsch 1949; Johnson 2003) describes positive and negative forms of interdependence in social systems. Positive interdependence is reflected in so-called promotive or collaborative interactions in which an aggregate within the system achieves its goal only to the extent to which the other aggregates achieve their goals as well. The negative interdependence is reflected in contrient or conflictual interactions in which aggregates only achieve their goals at the expense of other aggregates in the system. In line with social interdependence theory we argue that collaborative and conflictual interactions in the system as a whole have an impact on the interactions that unfold within the aggregates.

### 2.1. Summary and Definitions

In our analysis, we focus on the entwinement between expectations and experienced interactions and the way in which this entwinement transfers in the system in a bottom-up versus top-down

manner. We define *positive interdependence entwinement* as the association between expectations of positive interdependence and experienced collaboration (changes from expectations to what is really experienced after the interactions start) and *negative interdependence entwinement* as the association between expected negative interdependence and experienced conflict (changes from expected to experienced conflict). In what follows we will build on social interdependence theory to further specify the bottom-up and top-down influences in multiparty collaboration. Further on, we define *bottom-up influences* as the ways in which positive and negative entwinement transfers from stakeholder parties to the system as a whole. Finally, we define *top-down influences* as the ways in which the entwinement of positive and negative interdependence transfers from the large multiparty system to the stakeholder parties.

## 2.2. Bottom-Up Influences

Multiparty collaboration involves the exploration of various stakeholders' interests and working with the differences among the parties to identify or create common ground (Curşeu and Schruijer 2017; Schruijer and Vansina 2008). Collaboration does not imply the dissolution of stakeholders' boundaries, in other words, parties do not have to become the same, rather, they preserve their identity and find ways to achieve their own interests while working towards a joint goal. As parties develop relationships with other parties, they need to explore within their stakeholder party how to relate to the other stakeholder parties, where their points of connection are and where they could collaborate. Exploring possible collaboration with other parties is likely to stir up tensions within stakeholder parties as identities are challenged in confrontation with 'the other' and fears of being exploited, overruled or losing one's identity may be triggered. These tensions may result in frictions and conflict within the stakeholder parties. Relationship conflict refers to interpersonal frictions and has detrimental influences on group dynamics and group effectiveness while task conflict stands for addressing the different points of view openly and directly with the aim of arriving at a better way to deal with the task at hand based on an assessment and possible integration of these diverse views (Jehn 1995; Jehn et al. 1999; Curşeu and Schruijer 2017).

Bottom-up influences are the forces that originate in the relational dynamics experienced within the stakeholder parties while engaging in a collaborative process and have an impact on the intergroup dynamics at the multiparty system level. Conflict is a pervasive and multifaceted phenomenon in small groups (Jehn 1995), it is contagious and tends to spread among individuals and groups. For example, it has been shown that system-level conflicts around water use are often driven by social and cultural conflicts within stakeholder parties (Montaña et al. 2009). We therefore argue that changes in the relational frictions experienced within stakeholder parties tend to influence the conflictuality in the whole multiparty system. Likewise, changes in task conflict within stakeholder parties may have an impact on the collaborativeness in the whole system as positive experiences in constructively dealing with differences within one's stakeholder party may encourage representatives to engage in task conflict with representatives of the other stakeholder parties.

An illustrative example is presented in Bernard et al. (2014) related to sustainable agriculture. If farmers engage in continuous adaptation and change in order to meet their own needs they will be more likely to fulfil the expectations of their customers, investors and regulators and reduce the complaints of their neighbours. If, however farmers stick to the "old ways" of farming and resist change (lack of task conflict within the interest party) it is likely that their collaborativeness will decrease and the whole system will be composed of "loss-making investors, dissatisfied customers, angry neighbours and overacting regulators" (Bernard et al. 2014, p. 157). We posit a positive interplay between task conflict experienced within the stakeholder parties and the perceived collaborativeness of all other stakeholder parties in the multiparty system as a whole.

To summarize, we expect that the entwinement of positive and negative interdependence transfers from the individual stakeholder parties to the whole multiparty system (bottom-up influence). We therefore formulate the following specific hypotheses:

**Hypothesis 1.** *Changes from expected to experienced within-party relationship conflict lead to changes in the same direction from expected to experienced conflictuality among the parties in the system (bottom up transfer of negative interdependence entwinement).*

**Hypothesis 2.** *Changes from expected to experienced task conflict within parties lead to changes in the same direction from expected to experienced collaborativeness of all the other parties in the system (bottom up transfer of positive interdependence entwinement).*

*2.3. Top-Down Influences*

Top-down influences are created by the quality of the intergroup interactions that spiral down to influence the dynamics within the stakeholder parties. Based on open systems theory (Katz and Kahn 1978) and on a complex adaptive systems framework (Eidelson 1997) one can expect that forces at higher system levels limit the degrees of freedom at lower system levels.

Collaborativeness or promotive interactions reflect positive interdependence (Deutsch 1949; Johnson 2003). In the case of multiparty collaboration, promotive interactions are likely to exist when parties in the system realize that they need one another to realize their individual goals (Johnson 2003; Johnson and Johnson 2005). Through their individual actions, parties in the multiparty system promote the development and realization of collective and individual goals. As the collaborativeness of all stakeholder parties is perceived to increase, one may expect the tensions that are inherent to multiparty collaboration to decrease, which in turn may generate a positive relational climate within stakeholder parties. System collaborativeness creates more space for each stakeholder party to focus on the task and more courage may be shown to confront one another in the service of task accomplishment. As the tensions of multiparty interactions decrease and consume less time, more attention can be paid to the ideas and concerns of the individual members within parties (cf. Sherif and Sherif 1967); an increasing constructive climate at the system level may foster space to engage in constructive task conflict dealing with individual concerns at the party level. On the contrary, if the collaborativeness in the whole system decreases, the constructive engagement with the task at hand within stakeholder parties is likely to be jeopardized as anxieties regarding the other parties may distract the party members from the task at hand.

Conflictuality goes together with contrient or negative interdependence in multiparty systems. Negatively interdependent stakeholder parties realize their goals at the expense of goal realization of the other parties (Johnson 2003; Johnson and Johnson 2005). Parties engaging in conflictual interactions obstruct rather than support each other. Conflict is an emotionally laden experience (Pluut and Curşeu 2013) and tends to spiral down from the multiparty system level to the individual stakeholder parties. As stated in the social interdependence theory (Johnson and Johnson 2005) negative cathexis tends to spread among the participants in a system. Aligned with the emotional contagion model (Barsade and Gibson 1998), we expect that relational tensions experienced at the system level tend to induce relational frictions within stakeholder parties. One can imagine that when intergroup relational conflict is high, there is less attention to members' needs and concerns within a group (cf. Sherif and Sherif 1967), which in due time may cause frustration and friction internally.

We expect that the entwinement of positive and negative interdependence transfers from the multiparty system level to the stakeholder parties (top-down influence). We therefore hypothesize the following:

**Hypothesis 3.** *Changes from expected to experienced conflictuality perceived among the stakeholder parties in the multiparty system lead to changes in the same direction from expected to experienced relationship conflict within stakeholder parties.*

**Hypothesis 4.** *Changes from expected to experienced collaborativeness perceived among the stakeholder parties in the multiparty system lead to changes in the same direction from expected to experienced task conflict within the stakeholder parties.*

## 3. Methods

### 3.1. Simulation and Sample

One hundred and forty-five individuals (managers and consultants of which 54 women) with an average age of 43 years old participated in the study. Data was collected in five behavioural simulations that involved different participants. All participants were enrolled in postgraduate education at a business school or in a professional development program. We used a multi-party behavioural simulation (Schruijer and Vansina 2008; Vansina and Taillieu 1997; Vansina et al. 1998) in which seven or eight (depending on the number of participants) stakeholder parties engage in within-group as well as between-group interactions to deal with a complex regional development situation involving economic, social and environmental factors in the St Petersburg area, including the island Kotlin.

The participating parties were: a local authority (with an interest in the socio-economic situation on the island), a shipyard (the largest employer on the island facing a severe problem related to unemployment and decrease in business opportunities), an island-based yacht club (located on a scenic piece of land on the island and interested in developing their yacht club), a bank (interested in long-term investments), a group of young and rich entrepreneurs (with an emotional tie to the island), a Finnish yacht club (interested in new sailing routes for their members), a yacht club near St Petersburg (also wanting to expand their activities) and a technical school (associated with the Shipyard) (for more details see Vansina et al. 1998). Each party received the briefing describing their interests and had the freedom to develop their own strategy and approach.

At the onset of the simulation participants were allocated to a party based on their expressed preferences. Then each party received a booklet containing information concerning economic (e.g., risk of bankruptcy for the most important employer on the island), social (e.g., unemployment and social unrest) and environmental (e.g., water pollution) challenges in the region, as well as specific information concerning their own interests as a party.

The simulation proceeded in real time during a full day and few hours in the next morning and was followed by a joint debriefing. We allocated the first time slot of the simulation to the study of briefs within each party and develop their party strategy for interacting with the other stakeholder parties. No specific roles were assigned to the individual participants. After the within-party preparation, one-hour intergroup visiting and plenary meetings were alternated during the day. Intergroup visiting slots consisted of the possibilities to visit other parties (with a maximum of 3 parties being present at any particular time and place). During plenary meetings, all parties could be present and send their representative to the table, while the constituencies were allowed to sit behind their representatives and send notes.

The participating parties (36 groups having 3 to 4 members each over five simulations) were asked to fill out surveys concerning within-party (task and relationship conflict) as well as between-party dynamics (the perceived conflictuality and collaborativeness of all other stakeholder parties) three times during the simulation (before the interactions started, during interactions and after the simulation ended). In order to capture the entwinement between expectations and experienced social interactions, only the first and the second evaluations were used for further analyses as the intergroup interactions only started after filling out the first questionnaire.

### 3.2. Measures

**Task conflict** was evaluated with four items adapted from a conflict scale presented in Jehn (1995). For the first evaluation, at the onset of the simulation the content of the items were adapted to capture

expectations ("To what extent do you expect disagreements in your interest party related to the task?", "To what extent do you expect differences of opinion in your stakeholder party?", "How often do you think the members of your interest party will disagree about how things should be done?", "How often do you think the members of your interest party will disagree about which procedure should be used to do your work?"), while for the subsequent evaluation, the items referred to experienced task disagreements within one's stakeholder party (e.g., "To what extent are there disagreements in your stakeholder party that are related to the task," etc.). The answers were recorded on a 5-point Likert scale (1 to 5) and Cronbach's alpha for this scale was 0.94 at time 1 and .88 at time 2 for the group level aggregated items. In order to support aggregation, we used computed RWG (James et al. 1984) that ranged between 0.82 and 0.98 supporting the aggregation of the scores at the group level of analysis. The change from expectations of task conflict as evaluated before the interactions started to the experienced task conflict will therefore reflect the entwinement of positive interdependence at the stakeholder party level.

**Relationship conflict** was evaluated with four items adapted from the same conflict scale (Jehn 1995) and, like with the task conflict scale, at time 1 referring to expectations ("How much jealousy or rivalry do you expect to see among the members of your interest party?", "How often do you expect to have personality conflicts in your interest party?", "How much tension do you think will exist among the members of your stakeholder party?", "How often do you think people will get angry while working in your interest party?"). At time 2 the items referred to experienced interpersonal frictions (e.g., "How much jealousy or rivalry is there among the members of your stakeholder party," etc.). The same 5-point Likert scale was used (1 to 5) to record the answers. Cronbach's alpha for this scale was 0.95 at time 1 and .85 at time 2 for the items aggregated at the group level. The RWG scores ranged from 0.73 to 1.00 supporting the aggregation of the scores at the group level. The change from expected relationship conflict to experienced relationship conflict will therefore be indicative of the entwinement of negative interdependence at the stakeholder party level.

**Collaborativeness** was evaluated using a round robin technique in which each participant was asked to evaluate the extent to which he or she perceived each of the participating stakeholder parties as being collaborative. At time 1 participants were asked to estimate the collaborativeness of each stakeholder party based on the reading of the briefing material ("Please evaluate how collaborative you think each stakeholder party will be." The evaluations range from 0 = not collaborative at all to 5 = very collaborative"), while at time 2 they were asked to evaluate the experienced collaborativeness ("Please evaluate the extent to which [each stakeholder party] is collaborative." The evaluations range from 0 = not collaborative at all to 5 = very collaborative"). To capture bottom-up influences (what individual parties bring into the larger system), we aggregated evaluations for each stakeholder party as a referent—how each stakeholder party is perceived by the other stakeholder parties in the system. As an index of collaborativeness for bottom-up influences, we used the average score for the collaborativeness of each stakeholder party as perceived by the other stakeholder parties (total system-level score excluding one's own party)—that is the average collaborativeness ascribed to a particular stakeholder party by the others in the system. To capture top-down processes (how the multiparty interactions impact the within-party climate), we have aggregated evaluations using the whole system as a referent—how each stakeholder party perceives all the other stakeholder parties. As an index of collaborativeness for the top-down influences we have used the average score for how collaborative each party sees all the other stakeholder parties in the system (again excluding one's own party)—that is the collaborativeness ascribed to all other stakeholders in the system by a particular party. In other words, we have use collaborativeness of each stakeholder party as evaluated BY others as an index in the bottom-up analyses and collaborativeness attributed TO all the others by each stakeholder party as an index in the top-down analyses. Change from the expected to the experienced collaborativeness reflects the entwinement of positive interdependence and the two indices will be separately used to capture the bottom-up and top-down influences.

**Conflictuality** was evaluated in a similar fashion as collaborativeness, using a single item that referred to expected conflictuality of each party at time 1 ("Please evaluate how conflictual you think each party will be. The evaluations range from 0 = not conflictual at all to 5 = very conflictual") and evaluated experienced conflictual relations at time 2 ("Please evaluate the conflictuality of . . . [each of the other stakeholder parties]". The evaluations range from 0 = not conflictual at all to 5 = very conflictual"). To capture the bottom-up influences, a conflictuality score was computed by averaging all scores received by each stakeholder party from all the other stakeholder parties (total system-level score, excluding one's own party)—that is, conflictuality ascribed to a particular stakeholder party by all other parties in the system. To capture the top-down influences, we have used the average score for how each stakeholder party perceived the conflictuality of all the other stakeholder parties in the system (excluding one's own party)—that is conflictuality ascribed by a particular party to all the other parties in the system. We have therefore used conflictuality as evaluated BY others as an index in the bottom-up analyses and conflictuality in the system (ascribed TO others) as a whole for the top-down analyses. As for collaborativeness, the change from expected to experienced conflictuality indicates the entwinement of negative interdependence and the two indices are used to capture bottom-up and top-down influences.

To summarize, in order to capture the top-down and bottom up influences, we have used different aggregation procedures of the data collected through the round-robin procedure for collaborativeness and conflictuality. For the bottom-up influences the aggregation reflects what each party brings into the system (collaborativeness and conflictuality of each party as seen by others), while for the top-down influences, the aggregation reflects what each party perceives in the system as a whole (collaborativeness and conflictuality ascribed to others in the system).

## 4. Results

Means, standard deviations and correlations are presented in Table 1. In order to test the dynamic interplay between the two forms of conflict and the changing intergroup perceptions we used the MEMORE macro in SPSS version 22 (Montoya and Hayes 2017). The macro allows mediation tests in repeated measures designs and the procedure does not focus on the effect of a particular independent variable but rather models the effect of change induced by a particular event in a variable (mediator) on change in another variable (dependent variable). In our design, between the two successive evaluations, parties engaged in within and between-group interactions. This systematic change is considered as main independent variable. As we argued before, interactions within one's stakeholder party are expected to trigger changes in the conflict experienced between stakeholder parties, which in turn influences the way in which other stakeholder parties are perceived during intergroup interactions and vice versa. The macro is based on a bootstrapping procedure and it includes terms that capture change as well as the average scores of the mediator variables. Overall, as interactions in the simulation unfold, stakeholder parties seem to experience less within-group relationship conflict than originally expected (effect size = $-0.24$, $SE = 0.10$, $CI_{low} = -0.45$; $CI_{high} = -0.04$) and they are rated as less conflictual by others than initially expected (effect size = $-0.69$, $SE = 0.10$, $CI_{low} = -0.90$; $CI_{high} = -0.49$). In Hypothesis 1 we stated that as parties start interacting, changes in relationship conflict within stakeholder parties positively coevolve with changes in perceived conflictuality in the whole system. The results support this hypothesis as the indirect effect size is significant and the 95% confidence interval does not contain zero (indirect effect size = $-0.14$; $SE = 0.06$; $CI_{low} = -0.24$; $CI_{high} = -0.01$). The results of the analyses carried out with MEMORE are summarized in Figure 2 following the example reported in Montoya and Hayes (2017). As we expected, change in relationship conflict within stakeholder parties has a significant positive influence on change in perceived system conflictuality (effect size = 0.58; $SE = 0.17$ $CI_{low} = 0.24$; $CI_{high} = 0.91$), in other words, as relationship conflict experienced within stakeholder parties change, this change triggers changes in the perceptions of conflictuality at the system level.

**Table 1.** Presents the means, standard deviations and correlations among the variables used in further analyses.

| | Mean | SD | 1 | 2 | 3 | 4 | 5 | 6 | 7 | 8 | 9 | 10 | 11 |
|---|---|---|---|---|---|---|---|---|---|---|---|---|---|
| 1. TC Time1 | 2.88 | 0.56 | | | | | | | | | | | |
| 2. TC Time 2 | 2.64 | 0.40 | 0.09 | | | | | | | | | | |
| 3. RC Time 1 | 2.09 | 0.60 | 0.77 ** | 0.24 | | | | | | | | | |
| 4. RC Time 2 | 1.85 | 0.44 | 0.34 * | 0.51 ** | 0.35 * | | | | | | | | |
| 5. CollBY Time 1 | 3.01 | 0.33 | 0.02 | 0.21 | 0.16 | 0.18 | | | | | | | |
| 6. CollBY Time 2 | 2.33 | 0.51 | 0.16 | 0.65 ** | 0.33 * | 0.26 | 0.36 * | | | | | | |
| 7. ConflBY Time 1 | 2.57 | 0.58 | 0.45 ** | 0.12 | 0.60 ** | −0.02 | 0.01 | 0.17 | | | | | |
| 8. ConflBY Time 2 | 1.74 | 0.36 | −0.06 | −0.39 * | −0.24 | −0.09 | −0.10 | −0.58 ** | −0.08 | | | | |
| 9. CollTO Time 1 | 3.01 | 0.32 | 0.22 | 0.06 | 0.31 | 0.04 | 0.03 | −0.02 | −0.07 | −0.17 | | | |
| 10. CollTO Time 2 | 2.33 | 0.39 | 0.17 | 0.46 ** | 0.12 | 0.18 | 0.37 * | 0.40 * | 0.07 | −0.14 | 0.04 | | |
| 11. ConflTO Time 1 | 2.57 | 0.65 | 0.49 ** | 0.12 | 0.55 ** | −0.08 | 0.19 | 0.15 | 0.56 ** | −0.13 | 0.30 | 0.31 | |
| 12. ConflTO Time 2 | 1.74 | 0.48 | 0.03 | 0.16 | 0.15 | 0.09 | 0.14 | 0.10 | −0.12 | −0.02 | 0.16 | −0.06 | 0.16 |

Notes: * $p < 0.05$, ** $p < 0.01$; TC—task conflict within parties, RC—relationship conflict within parties, CollBY—collaborativeness ascribed by others, ConflBY—conflictuality ascribed by others, ColTO—collaborativeness ascribed to others, ConflTO—conflictuality ascribed to others.

Overall, as interactions unfold in the simulations, parties seem to experience less task conflict within stakeholder parties than originally expected (effect size = −0.23; *SE* = 0.11, $CI_{low}$ = −0.46; $CI_{high}$ = −0.01) and stakeholder parties are perceived as being less collaborative by others than initially expected (effect size = −0.61; *SE* = 0.08, $CI_{low}$ = −0.77; $CI_{high}$ = −0.46). Hypothesis 2 stated that as intergroup interactions begin, changes in task conflict within stakeholder parties positively coevolve with the changes in perceived system collaborativeness. Although the results do not directly support the mediation hypothesis as the 95% confidence intervals for the indirect effect include zero (indirect effect size = −0.06; *SE* = 0.04; $CI_{low}$ = −0.14; $CI_{high}$ = 0.01), the direct effect of change in task on conflict within stakeholder parties on change on perceived system collaborativeness is positive and significant (effect size = 0.26; *SE* = 0.11 $CI_{low}$ = 0.03; $CI_{high}$ = 0.50). In other words, change in perceived system collaborativeness is directly and positively triggered by changes in task conflict within stakeholder parties. Moreover, as also illustrated in Figure 3 the change in collaborativeness is also positively predicted by the average task conflict in the two evaluation points. We can therefore conclude that Hypothesis 2 received partial support.

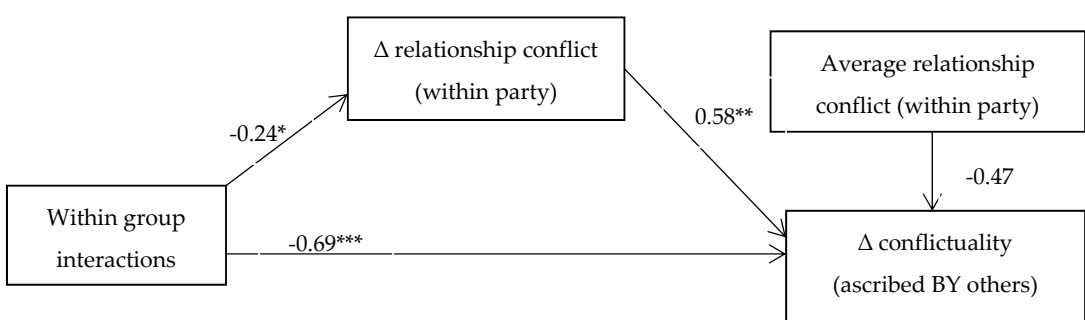

**Figure 2.** The overall mediation model for change in relationship conflict and conflictuality (bottom-up influence). Notes: * $p < 0.05$; ** $p < 0.01$, *** $p < 0.001$, Δrelationship conflict = relationship conflict at time 2 minus relationship conflict at time 1 and describes the negative interdependence entwinement at the stakeholder party level, Δconflictuality = conflictuality as perceived by others at time 2 minus conflictuality as perceived by others at time 1 and reflects the negative interdependence entwinement at the multiparty level.

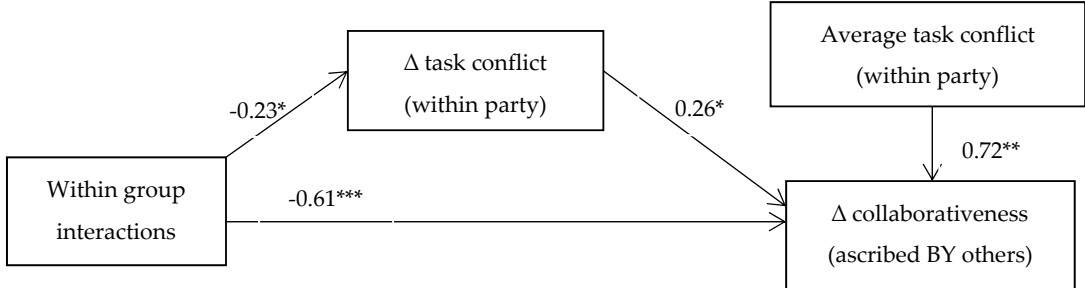

**Figure 3.** The overall mediation model for change in tach conflict and collaborativeness (bottom-up influence). Notes: * *p* < 0.05; ** *p* < 0.01, *** *p* < 0.001, Δtask conflict = task conflict at time 2 minus task conflict at time 1 (positive interdependence entwinement at the stakeholder party level), Δcollaborativeness = collaborativeness as perceived by others at time 2 minus collaborativeness as perceived by others at time 1 (positive interdependence entwinement at the multiparty level).

As interactions started during the simulation, parties seem to perceive the others as less conflictual than they expected initially (effect size = −0.83; *SE* = 0.12; *CI*$_{low}$ = −10.09; *CI*$_{high}$ = −0.58). Hypothesis 3 stated that changes in conflictuality at the system level are predictive of changes of changes in relationship conflict experienced within stakeholder parties. The mediation results support the mediation role for system conflictuality as the estimated confidence interval for the indirect effect does not include zero (indirect effect size = −0.23; *SE* = 0.12; *CI*$_{low}$ = −0.49; *CI*$_{high}$ = −0.01), the direct relation between change in conflictuality on change in experienced relationship conflict within parties is positive and significant (effect size = 0.28; *SE* = 0.12 *CI*$_{low}$ = 0.04; *CI*$_{high}$ = 0.52) supporting a top-down effect of system conflictuality on within party relationship conflict. The overall results supporting Hypothesis 3 are presented in Figure 4.

After interactions started during the simulation, parties perceive the others as less collaborative than expected (effect size = −0.67; *SE* = 0.08; *CI*$_{low}$ = −0.84; *CI*$_{high}$ = 0.51). Hypothesis 4 stated that the changes in collaborativeness at the system level co-evolve with changes in task conflict experienced within stakeholder parties. The results of this mediation analyses are presented in Figure 5 and reveal no significant indirect effect as the estimated confidence interval include zero (indirect effect size = −0.18; *SE* = 0.18; *CI*$_{low}$ = −0.60; *CI*$_{high}$ = 0.11) and the direct association between change in collaborativeness at the system level and task conflict experienced within parties is also not significant (effect size = 0.27; *SE* = 0.23 *CI*$_{low}$ = −0.20; *CI*$_{high}$ = 0.75).

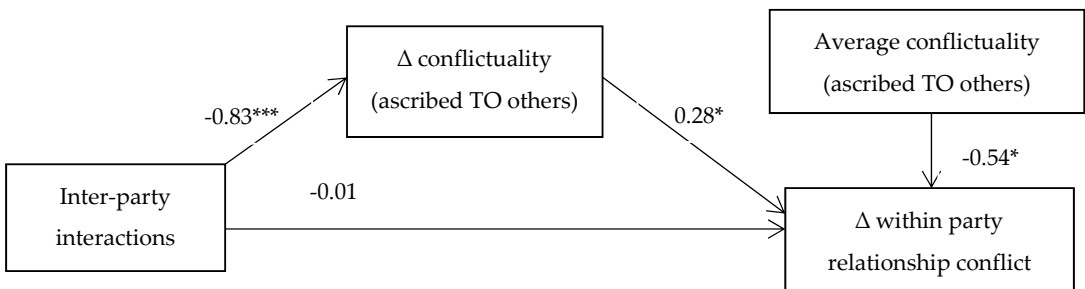

**Figure 4.** The overall mediation model for change in conflictuality and relationship conflict (top-down influence). Notes: * *p* < 0.05; *** *p* < 0.001, Δrelationship conflict = relationship conflict at time 2 minus relationship conflict at time 1, Δconflictuality = the conflictuality as perceived by each party for the system as a whole at time 2 minus conflictuality at the system level at time 1.

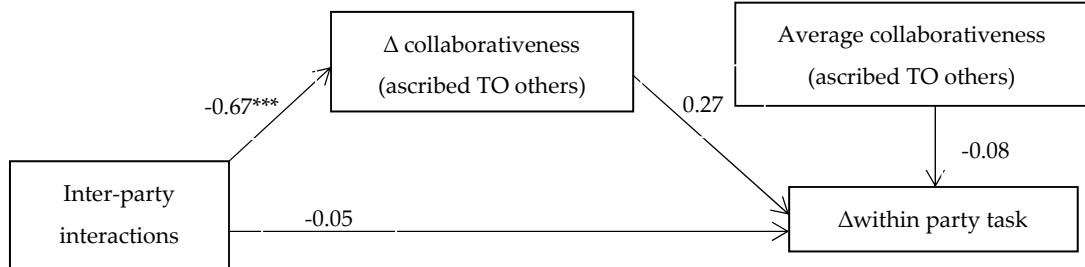

**Figure 5.** The overall mediation model for change in tach conflict and collaborativeness (top down influence). Notes: *** *p* < 0.001, Δtask conflict = task conflict at time 2 minus task conflict at time 1, Δcollaborativeness = collaborativeness as perceived by each party for the system as a whole at time 2 minus collaborativeness as for the system as a whole at time 1.

## 5. Discussion

Our paper addressed the interrelatedness of multiparty dynamics at different system levels. We focused on the entwinement between expectations and social interactions and analysed positive and negative interdependence (the interplay between collaboration and conflict). Moreover, we tested bottom-up and top-down influences, that is, how within- stakeholder party dynamics affect those at the system level, and, how system level dynamics affect dynamics at the stakeholder level, in a multiparty context. Building on social interdependence theory (Deutsch 1949; Johnson 2003) we argued that positive interdependence (facilitating collaborativeness) at the system level increase the degrees of freedom for elemental components of the system (allowing stakeholder parties to engage with the task and thus engage in within-group task conflict), while negative interdependence (facilitating conflictuality) decreases these degrees of freedom and stimulate relationship conflict within stakeholder parties.

As top-down influences we have tested the effect of the changes in collaborativeness and conflictuality perceived at the system level on the changes in task and relationship conflict experienced within stakeholder parties. Regarding bottom-up influences, we have tested the extent to which changes in task and relationship conflicts experienced within the stakeholder parties co-evolve with changes in collaborativeness and conflictuality at the system level. To disentangle the top-down and bottom-up influences we have used different referential models for collaboration and conflict. Overall, bottom-up processes received stronger support than top-down processes. Changes in task conflict within stakeholder parties are positively related to changes in perceived collaborativeness at the system level (that is, perceived collaborativeness of all stakeholder parties combined) while changes in within-group relationship conflict are positively associated with system conflictuality. From the two top-down influences investigated, only the effect of change in conflictuality at the system level on the changes in relationship conflict experienced within stakeholder parties was significant. This finding next to the significant bottom-up association between relationship conflict and perceived conflictuality supports the pervasive nature of especially relationship conflict in social systems (Van Bunderen et al. 2017; Pluut and Curşeu 2013). Conflict contagion or the transfer of negative cathexis (Johnson and Johnson 2005) seems to be an influential process in multiparty systems. Social interdependence theory (Deutsch 1949; Johnson and Johnson 2005) also claims that positive cathexis could be transferred among the parties engaged in the system. This claim warrants further empirical exploration as positive cathexis is claimed to be beneficial for collaboration (Johnson and Johnson 2005).

The cognitive synergy path (conceptually described in Curşeu and Schruijer 2017) and the positive interplay between changes in within-party task conflict and system collaborativeness are only supported for the bottom-up test. Changes in task related debates, in this case engaging in task conflict, within the stakeholder parties, are proportionally related to changes in parties' perceived collaborativeness. In other words, the more stakeholder parties engage internally with the different opinions regarding addressing the task, the more they have the potential to contribute to the

overall collaboration in the system. Using the terms of social interdependence theory, task conflict seems to increase the perceived inducibility (openness to being influenced by others) of the parties. As inducibility increases, parties engage with the multiparty task at hand and promote the achievement of their collective goal (Johnson and Johnson 2005).

We call for more research integrating systems-psychodynamic theorizing (Schruijer 2016; Vansina and Vansina-Cobbaert 2008) and social interdependence theory (Johnson and Johnson 2005) as they both apply to multiparty systems. In social interdependence terms, for multiparty collaboration *substitutability* needs to be low (parties have unique competencies or resources that cannot be substituted by the other parties in the system), while *inducibility* is required (parties need to be open to social influence in the process of developing a collective goal) and *cathexis* is present (the transfer of positive and negative evaluations/emotions within and among parties). A key claim of social interdependence theory is that the structure of the individual goals in a given situation determines how participants interact. The structure of the stakeholder parties' goals in our simulation require collaboration or promotive interactions as parties cannot deal with the complex situation on their own. However, as our results show, parties often engage in conflictual (contrient) interactions and the emotional dynamics override the normative (cognitive and rational) need for collaboration. Exploring the (conscious and unconscious) emotional dynamics or the cathexis using a systems psychodynamic perspective could help us gain a deeper understanding of the dynamics of multiparty systems.

So far, we have offered a system interpretation of our results. In our simulation, different individuals engaged in interactions as representatives of the seven parties. They had to embark on these interactions without the restrictions of particular (individual) role prescriptions. As such, their interpersonal skills, personality and behavioural tendencies may have played an important role on how the interactions between group members unfolded. The aim of the simulation was not to attend to individual behaviours and no personal feedback was given to participants. Moreover, as the participants knew each other (they all took part in either a degree program lasting for 16 months or a professional development program), interpersonal histories may have been carried into the simulation and interpersonal conflicts from outside the simulation or interpersonal attraction for that matter, could have influenced the inter-group dynamics.

Our paper has several limitations. First, for the test of top down influences, data to test the models were collected from the same source for the mediator and dependent variables, therefore these analyses are susceptible to common source bias. However, given the fact that the evaluation of within-party conflict and the conflictuality and collaborativeness had different reference points and different rating intervals, we could argue that the common method bias concerns are attenuated. Second, in our study we only evaluated task and relationship conflict as experienced within the stakeholder parties and future research could also evaluate task and relationship conflict in the whole system and in this way test top-down and bottom-up influences in a different manner. Third, we have used aggregated individual scores to obtain evaluations of system collaborativeness and conflictuality. Future research could use global evaluations made by external observers for these two forms of social interdependence. Fourth, in terms of alternative interpretations, one could argue that the bottom-up influences as operationalized in our study actually reflect self-fulfilling prophecies (Merton 1948). Participants' expectations about positive and negative interdependence may lead them to behave in ways that are consistent with these expectations and as a consequence the way they are perceived by others in interactions is congruent with their own initial expectations. Moreover, the way we have operationalized the top-down influences may conflate false consensus effects (Ross et al. 1977). In other words, participants' experiences of positive or negative interdependence in their own parties may lead them to evaluate other parties as displaying the same forms of interdependence. Although, based on the data collected in this study we cannot refute these two alternative explanations, we argue that the aggregation of individual evaluations may have alleviated the self-fulfilling prophecy and false consensus influences. Finally, given the time-intensive nature of the behavioural simulations,

our sample is small and future research could attempt to replicate our findings in larger samples and in different settings.

*Practical Implications*

Due to its pervasive negative influences, relationship conflict needs to be pro-actively managed in multiparty systems. Conflict reducing techniques both at the party as well as at the system level may help alleviate the detrimental influences of conflict escalation. Interventions aimed at generating task conflict within stakeholder parties will spur collaborativeness in the whole system. Simple normative interventions could help stakeholder parties to engage in healthy task debates as illustrated by research that used ground rules for collaboration to increase the quality of group debates and ultimately group rationality (Curşeu and Schruijer 2012). Moreover, process consultation may support multiparty collaboration by stimulating healthy task conflict and reducing the relationship conflict. Finally, behavioural simulations constitute experiential learning opportunities in which individuals can gain an understanding of multiparty dynamics and can develop their collaboration skills.

**Author Contributions:** S.S. and P.L.C. designed and conducted the study; P.L.C. analysed the data; S.S. provided the behavioural simulation materials and brochures; P.L.C. and S.S. wrote and revised the paper.

**Funding:** P.L.C. was supported by a grant of the Romanian National Authority for Scientific Research, CNCS—UEFISCDI, project number PN-III-P4-ID-ERC- 2016-0008. The funders had no role in study design, data collection and analysis, decision to publish, or preparation of the manuscript.

**Conflicts of Interest:** The authors declare no conflicts of interest.

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
