# Peer review of "Cross-Level Dynamics of Collaboration and Conflict in Multi-Party Systems: An Empirical Investigation Using a Behavioural Simulation"

_admsci, doi:10.3390/admsci8030026_

Round 1

Reviewer 1 Report

I enjoyed reading this piece, and think that it offers a few relatively new and interesting contributions. First, the focus on both within-and between-group interactions is really useful and relatively underexplored in collaboration literature. Second, the methodology is a refreshing change from the case study research that comprises the bulk of the collaboration research.

I do find the reference list kind of unusual - it’s a little bit all over the place with respect to different disciplines, and a good amount of the literature is surprisingly old. As a result, the piece feels very current with respect to the methodology but not really tapping into the current debates in the cross-disciplinary collaboration literature. I’d encourage the authors to include more contemporary references to this literature and include illustrative examples that better relate to this literature in their lit review.

The hypotheses, as written, are rather unusual – I’d expect to see something more concise and direct, without explanation or qualifiers such as “we expect.” If you’re outlining the theory properly in the literature review, then the hypotheses – which test the theory – should be relatively concise statements that don’t require additional explanation or alternative wordings. See other examples in ASQ (after reading the results section and then returning back to the front end of the paper, I notice that your hypotheses are stated much more clearly in the results).

In Methods, please clarify whether “stakeholder parties” means that the individuals representing participating each represent groups of people, or whether stakeholder parties also refers to individuals? Were the stakeholder parties further specified?

Overall, I think the paper suffers a bit from a lack of clarification on what constitutes a stakeholder – the paper begins by emphasizing multiple organizations, but then doesn’t distinguish between individuals and organizations in terms of what the roles are within the behavioral simulation. This should be clearer so that we can fully understand the rest of the measures – is this referring to individual relationships or relationships between organizations? This is a common concern within collaboration literature and needs to be addressed here.

In Methods, appreciated the distinctions between task and relationship conflict.

Did I miss the definitions for collaborativeness and conflictuality? Collaboration is defined in many different ways by researchers across disciplines (and for scholars of interorganizational collaboration, there are different typologies for within-sector and cross-sector collaboration), so further definition here would be helpful and can better link to the collaboration literature. I’d like to see more here on defining conflictuality as well.

In the findings and discussion, I again am wondering about the distinction between individuals and organizations and wondering how to best interpret your results – as individuals participants engaged in dynamics with other individuals, as individuals participating on behalf of organizations but referring largely to individual conflicts, or as individuals participating on behalf of organizations but referring more to task and relationship conflict related to organizational representation? Overall, I’d like to see a better distinction between individuals, groups, and organizations – this will help strengthen the contribution of the paper as well as tie in the appropriate references.

I find the findings very interesting – the emphasis on bottom-up dynamics are really interesting. I’d like to see more ramifications and applications of this – but the concluding sections are so short! Can you elaborate more on some of these findings and tie them into the broader collaboration literature?

Appreciate the inclusion of limitations here, but I might also mention, since this paper deals with a behavioral simulation, the acknowledgment that conflict is different in zero-history groups than it is in bona fide groups. Did any of the participants know each other beforehand? This also isn’t clear in the methodology but might be relevant in studies involving conflict or collaboration.

Thanks for the opportunity to read your paper, and I wish you the best with the manuscript.

Author Response

Answer: Thank you for your appreciative remarks and for the suggestions to improve our manuscript.We provide an itemized reply to your comments in red (under Answer) to each of your remarks (in black).

 I do find the reference list kind of unusual - it’s a little bit all over the place with respect to different disciplines, and a good amount of the literature is surprisingly old. As a result, the piece feels very current with respect to the methodology but not really tapping into the current debates in the cross-disciplinary collaboration literature. I’d encourage the authors to include more contemporary references to this literature and include illustrative examples that better relate to this literature in their lit review.

Answer: thank you for pointing our this shortcomings. We agree with the fact that the theoretical framework is eclectic, yet to some extent this is tied to the variety of approaches in collaboration research. Multi-party collaboration was explored and theorized by researchers in a variety of disciplines from psychology to public administration. We have revised the theoretical framework and excluded some of the references to the complex adaptive systems that might have been confusing for the reader. In the revised version of the manuscript we focused more on the Social Interdependence Theory as a core theoretical framework. We also bring in more insights from the collaboration literature to support this framework – in particular the practice oriented theory of collaboration (Huxham & Vangen) and the systemic approaches to multi-party collaboration (Wood & Gray, 1991).

The hypotheses, as written, are rather unusual – I’d expect to see something more concise and direct, without explanation or qualifiers such as “we expect.” If you’re outlining the theory properly in the literature review, then the hypotheses – which test the theory – should be relatively concise statements that don’t require additional explanation or alternative wordings. See other examples in ASQ (after reading the results section and then returning back to the front end of the paper, I notice that your hypotheses are stated much more clearly in the results).

Answer: we have rephrased the hypotheses and we hope they are clearer in the current version of the manuscript. Moreover, in the section on summary and definitions, we tie more directly the concepts of entanglement to the social interdependence theory concepts and arguments.

 In Methods, please clarify whether “stakeholder parties” means that the individuals representing participating each represent groups of people, or whether stakeholder parties also refers to individuals? Were the stakeholder parties further specified? Overall, I think the paper suffers a bit from a lack of clarification on what constitutes a stakeholder – the paper begins by emphasizing multiple organizations, but then doesn’t distinguish between individuals and organizations in terms of what the roles are within the behavioral simulation. This should be clearer so that we can fully understand the rest of the measures – is this referring to individual relationships or relationships between organizations? This is a common concern within collaboration literature and needs to be addressed here.

Answer: in line with your and the other reviewer’s remarks, we have revised the methods section and added more information on the simulation. Also we included more details on how the groups were formed and the types of interactions that took place during the simulation. Moreover, we reflect on the nature of the task in line with the generic view of multi-party collaboration. We have provided a short description of the participating parties and clarified that they had the freedom to decide their own approach and strategy. We have also clarified that no specific roles were assigned to the individual participants, except from the roles they defined for themselves after discussing with their own party members. So the simulation really taps into the complexity of inter-organizational collaboration.

In Methods, appreciated the distinctions between task and relationship conflict.

Did I miss the definitions for collaborativeness and conflictuality? Collaboration is defined in many different ways by researchers across disciplines (and for scholars of interorganizational collaboration, there are different typologies for within-sector and cross-sector collaboration), so further definition here would be helpful and can better link to the collaboration literature. I’d like to see more here on defining conflictuality as well.

Answer: in line with your and the other reviewer’s remarks we clarify better how the measures were derived, how the data was aggregated and we now clearly specify the way in which the two measures of conflictuality and collaborativeness are related to the bottom-up and top-down influences.

 In the findings and discussion, I again am wondering about the distinction between individuals and organizations and wondering how to best interpret your results – as individuals participants engaged in dynamics with other individuals, as individuals participating on behalf of organizations but referring largely to individual conflicts, or as individuals participating on behalf of organizations but referring more to task and relationship conflict related to organizational representation? Overall, I’d like to see a better distinction between individuals, groups, and organizations – this will help strengthen the contribution of the paper as well as tie in the appropriate references.

Answer: the way in which the simulation is designed  gives this behavioral simulation an inter-organizational focus. No specific roles were assigned to the individual members and no individual feedback was given during the debriefing. Therefore the whole focus of the simulation was on the emerging dynamics of inter-organizational relationships. The way in which we have aggregated the data also reflect this level of analysis. We clarify now in the discussion that we have focused on the systemic dynamics and not on interpersonal dynamics. We also acknowledge the role of pre-existing interpersonal ties might have played in the simulation and we also acknowledge that we did not explicitly evaluate these possible influences in the simulation. One of the new paragraphs reads: “So far, we have offered a systemic interpretation of our results. In our simulation,  different individuals engaged in interactions as representatives of the seven parties. They had to embark on these interactions without the restrictions of particular (individual) role prescriptions. As such, their interpersonal skills, personality and behavioral tendencies may have played an important role on how the  interactions between group members unfolded. The aim of the simulation was not to attend to individual behaviors and no personal feedback was given to participants. Moreover, as the participants knew each other (they all took part in either a degree program or a professional development program), interpersonal histories may have been carried into the simulation and interpersonal conflicts from outside the simulation or interpersonal attraction for that matter could have influenced the inter-group dynamics.  

I find the findings very interesting – the emphasis on bottom-up dynamics are really interesting. I’d like to see more ramifications and applications of this – but the concluding sections are so short! Can you elaborate more on some of these findings and tie them into the broader collaboration literature?

Answer: Given space constraints we have decided not to follow up on this request and refer the reader to the introduction section where we spell out how we think we contribute to the larger literature. We have revised the intro and added, as mentioned in our reply to an earlier remark, some more general theoretical approaches in the collaboration literature.

Appreciate the inclusion of limitations here, but I might also mention, since this paper deals with a behavioral simulation, the acknowledgment that conflict is different in zero-history groups than it is in bona fide groups. Did any of the participants know each other beforehand? This also isn’t clear in the methodology but might be relevant in studies involving conflict or collaboration.

Answer: We have offered additional information on participants and the way they were distributed to the parties. We also acknowledge that substantively the nature of conflict did vary across the simulations, yet we were interested in the magnitude of conflict and the systemic dynamics of conflict and collaboration. We hope we managed to clarify this issue in the revision.

Reviewer 2 Report

The manuscript analyses how expected and perceived intra-group and inter-group levels of conflict and cooperation are related in a behavioral simulation. The authors have picked an interesting problem that is both timely and important. To study the problem the authors performed a behavioral simulation study with the participation of 145 managers and consultants. The use of a behavioral simulation is likely to reveal more than pure observation or survey methods and the use of managers and consultants is likely to produce results that are generalizable to a business context. 

The authors hypothesize both bottom-up and top-down influences in the multiparty system being studied and use self-reported data from the participants to perform mediation analyses in order to investigate their hypotheses.

The authors draw on many different theoretical frameworks resulting in a rather eclectic overall impression. This in itself need not be a problem, but it is a challenge to both the authors and the readers, and in my view the authors have an obligation to tie the many strands into a coherent whole and present this to the reader in a compelling way. In this regard the present manuscript falls short.

In my opinion the two biggest problems with the current manuscript is (i) that there are too few details about the actual procedure and (ii) too little reporting of the empirical data to properly evaluate the quality of the research. I elaborate on these two issues in the following.

For an article reporting on a study to be accepted it should be possible for other researchers reading the article to reproduce the study. This is not possible based on the description in the current manuscript. The authors should provide a detailed account of the behavioral simulation, possibly in supplementary material, or a reference to the particular task used. It is not clear if the task is provided in one of the references provided (Schruijer & Vansina, 2008; Vansina & Taillieu, 1997; Tallieu & Schruijer, 1998).

Regarding the data, there are two issues: First, the definition of each measure should be clear, e.g. all the questions used in questionnaires. As an example the authors state (p. 6) that “Task conflict was measured with four items adapted from a conflict scale presented in Jehn (1995).” But which four, why those, and how were they adapted? If the scale in Jehn (1995) has been validated, then why not just adapt it as is? On p.8 the authors write “In our design, between the two successive evaluations, groups engaged in within and between group interactions. This systematic change is considered as main independent variable.” Even after reading it several times, I still have no idea how exactly this variable is defined. It is not clear to me what the systematic change is — probably because the task has not been described in detail. This is much too vague. In the mediation models this variable is called “within group interaction” and inter-group interaction” but again I could not find a concise definition of these. The second issue is that there is no description or presentation of the actual data in the manuscript. What are the means and standard deviations of the variables measured at different times in the simulation? What is the correlation between different variables? The authors never really present their data, only the results of analyzing them. This makes the manuscript very opaque, and I would urge the authors to clearly present their data (and perhaps making the data set available online).

Another problem is that it is difficult to find the connection between the many theoretical frameworks brought into play and the way the data are analyzed. As an example, the introduction pays homage to Complex Adaptive Systems (CAS) and mentions that both bottom-up and top-down mechanisms are at play simultaneously. But when the data are analyzed (e.g. collaborativeness) the bottom-up and top-down measures are based on the same data (but aggregated in different ways). This seems at odds with conventional CAS approaches, where an emergent property is seen as qualitatively different from the micro-behavior from which it arises. Here the two seem very similar, indeed almost conflated. There are also concepts, such as entanglement, that are used in the manuscript, but not related in a substantial way to the data or analysis. This may only be sloppy language, but it leaves a somewhat fuzzy overall impression.

In view of the problems already described, I see no reason to go into details regarding the analysis and discussion sections of the manuscript.

As a general comment the language in the manuscript seems overly complex and convoluted, and I sincerely believe that the quality will improve dramatically if the authors describe their work in a simpler and more direct way.

I realize that this review may come across as very negative, but my intention is to provide actionable comments, and I hope that the authors can use my review in their further work. Research is a complex endeavor and it is not always easy to convey the results. My guess is that this work has been submitted without the authors seeking feedback, either by presenting it at a seminar or by getting a friendly review by a colleague. I would recommend the authors to seek out such feedback.

I like the empirical approach taken by the authors but would have liked to see an experimental manipulation as part of the design, to alleviate endogeneity. It would also have strengthened the study design if there were behavioral measures, rather than relying on self-report.

Author Response

Answer: Thank you for your appreciative remarks and for the suggestions to improve our manuscript. In red, we provide an itemized reply to your comments (in black).

The authors draw on many different theoretical frameworks resulting in a rather eclectic overall impression. This in itself need not be a problem, but it is a challenge to both the authors and the readers, and in my view the authors have an obligation to tie the many strands into a coherent whole and present this to the reader in a compelling way. In this regard the present manuscript falls short.

Answer: thank you for pointing our this shortcomings. We agree with the fact that the theoretical framework is eclectic, yet to some extent this is tied to the variety of approaches in collaboration research. Multi-party collaboration was explored and theorized by researchers in a variety of disciplines from psychology to public administration. We have revised the theoretical framework and excluded some of the references to the complex adaptive systems that might have been confusing for the reader. In the revised version of the manuscript we focused more on the Social Interdependence Theory as a core theoretical framework. We also bring in more insights from the collaboration literature to support this framework – in particular the practice oriented theory of collaboration (Huxham & Vangen) and the systemic approaches to multi-party collaboration (Wood & Gray, 1991).

In my opinion the two biggest problems with the current manuscript is (i) that there are too few details about the actual procedure and (ii) too little reporting of the empirical data to properly evaluate the quality of the research. I elaborate on these two issues in the following.For an article reporting on a study to be accepted it should be possible for other researchers reading the article to reproduce the study. This is not possible based on the description in the current manuscript. The authors should provide a detailed account of the behavioral simulation, possibly in supplementary material, or a reference to the particular task used. It is not clear if the task is provided in one of the references provided (Schruijer & Vansina, 2008; Vansina & Taillieu, 1997; Tallieu & Schruijer, 1998).

Answer: we acknowledge the relevance of replicability in social sciences. We also acknowledge the contextual nature of most collaborative relationships. The behavioral simulation described in the paper is designed as an experiential learning tool, that allows participants to experience collaboration in a realistic setting. The task therefore is open and true to the various definitions of collaboration (“parties who see different aspects of a problem can constructively explore their differences and search for solutions that go beyond their own limited vision of what is possible” Gray 1989, p. 5”, or “Collaboration is a process in which autonomous actors interact through formal and informal negotiation, jointly creating rules and structures governing their relationships and ways to act or decide on the issues that brought them together;” Thomson & Perry, 2006, p. 23). So there is no pre-defined task in this behavioral simulation. The parties are brought together in view of their potential goal interdependencies yet the way in which they will work with their similarities and differences is not pre-defined nor constrained by a particular task structure. They are invited to use the time available whatever way they find appropriate: builds relationships across boundaries, collaborate, compete, or sit in their own room all day. The structuring of the situation is minimalistic, in that the participants are communicated the ground rules for interactions and the dual organization of intergroup interactions – plenary meetings through representatives and visiting in which no more than 3 groups could be present at the same time. In the paper we have provided more  details on the simulation and also have clarified the minimal task structure that was introduced in the system.

Regarding the data, there are two issues: First, the definition of each measure should be clear, e.g. all the questions used in questionnaires. As an example the authors state (p. 6) that “Task conflict was measured with four items adapted from a conflict scale presented in Jehn (1995).” But which four, why those, and how were they adapted? If the scale in Jehn (1995) has been validated, then why not just adapt it as is? On p.8 the authors write “In our design, between the two successive evaluations, groups engaged in within and between group interactions. This systematic change is considered as main independent variable.” Even after reading it several times, I still have no idea how exactly this variable is defined. It is not clear to me what the systematic change is — probably because the task has not been described in detail. This is much too vague. In the mediation models this variable is called “within group interaction” and inter-group interaction” but again I could not find a concise definition of these. The second issue is that there is no description or presentation of the actual data in the manuscript. What are the means and standard deviations of the variables measured at different times in the simulation? What is the correlation between different variables? The authors never really present their data, only the results of analyzing them. This makes the manuscript very opaque, and I would urge the authors to clearly present their data (and perhaps making the data set available online).

Answer: we have added all items used in the questionnaire and clarified that the adaptation refers to the evaluations of expectations (before the simulation) and the real interactions (during the simulation). Moreover, we present the means, standard deviations and correlations in Table 1. As we defined in the section on “Summary and definitions” the entanglement refers to the association between the expectations and real interpersonal experiences so it is estimated by the mediation model used in our paper. As stated in the paper introducing the method for estimating mediation in repeated measures models, the independent variable refers to the systematic change that occurred between the two measurement points. For our sample this systematic change refers to the onset of the interpersonal interactions. As such, in line with the social interdependence theory we estimate the extent to which the changes induced by the onset of interpersonal interactions in a particular variable (group level or systemic depending on the type of influence we modeled) leads to changes in other variables. This change we labeled entanglement – in line with the arguments derived from the social interdependence theory that expectations of positive/negative interdependence are linked with the types of interpersonal interactions experienced later on.

Another problem is that it is difficult to find the connection between the many theoretical frameworks brought into play and the way the data are analyzed. As an example, the introduction pays homage to Complex Adaptive Systems (CAS) and mentions that both bottom-up and top-down mechanisms are at play simultaneously. But when the data are analyzed (e.g. collaborativeness) the bottom-up and top-down measures are based on the same data (but aggregated in different ways). This seems at odds with conventional CAS approaches, where an emergent property is seen as qualitatively different from the micro-behavior from which it arises. Here the two seem very similar, indeed almost conflated. There are also concepts, such as entanglement, that are used in the manuscript, but not related in a substantial way to the data or analysis. This may only be sloppy language, but it leaves a somewhat fuzzy overall impression.

Answer: thank you for pointing out the theoretical issues. We have excluded the focus on CAS, yet we added some more recent theoretical approaches on collaboration as also suggested by the second reviewer. Now the paper focuses on the social interdependence theory as a key theoretical framework and we define the terms used in the paper in line with this theory – including the entanglement of positive and negative interdependence. Entanglement will refer to the connection between expectations and interpersonal interactions, a key element in the social interdependence theory. With respect to the data aggregation, the way in which we aggregated the data fits the way in which we have theorized the bottom-up and top down influences. For the bottom-up influences the aggregation reflects what each team brings into the system (collaborativeveness and conflictuality of the party as seen by others), while for the top-down influences, the aggregation reflects what each party sees in the system (collaborativeness and conflictuality ascribed to others). The task and relationship conflict were aggregated at the group level of analysis as the items used in these scales refer to group level dynamics. We added more insights and clarifications on these aggregation procedures and we hope things are more clear in the current version of the manuscript. We understand your concern with the conflation of scores, yet although we derived the scores from the same round robin procedure, the scores reflect different dynamics and are aligned with the way we have theorized the top-down and bottom-up influences. The correlations among these scores excludes the interpretation that these scores are conflated.    

In view of the problems already described, I see no reason to go into details regarding the analysis and discussion sections of the manuscript.

As a general comment the language in the manuscript seems overly complex and convoluted, and I sincerely believe that the quality will improve dramatically if the authors describe their work in a simpler and more direct way.

I realize that this review may come across as very negative, but my intention is to provide actionable comments, and I hope that the authors can use my review in their further work. Research is a complex endeavor and it is not always easy to convey the results. My guess is that this work has been submitted without the authors seeking feedback, either by presenting it at a seminar or by getting a friendly review by a colleague. I would recommend the authors to seek out such feedback.

I like the empirical approach taken by the authors but would have liked to see an experimental manipulation as part of the design, to alleviate endogeneity. It would also have strengthened the study design if there were behavioral measures, rather than relying on self-report.

Answer: thank you for your constructive remarks on our manuscript. We have worked with your and the other reviewer’s comment and thoroughly revised the manuscript. We believe that the paper is much improved and we hope the message conveyed is clearer in the current version of the manuscript.

Round 2

Reviewer 1 Report

This version is more appropriately grounded in the collaboration literature and addresses the complexity and paradox of this body of research. Overall, the theoretical framework is better specified in this version, and the addition of “entwinement” is helpful here. The hypotheses and findings are better explained, and the merits and limitations of the experiment are much clearer in this revision.